# Effect of Multimorbidity on Fragility Fractures in Community-Dwelling Older Adults: Shimane CoHRE Study

**DOI:** 10.3390/jcm10153225

**Published:** 2021-07-22

**Authors:** Garu A, Shozo Yano, Abdullah Md Sheik, Aorigele Yu, Kenta Okuyama, Miwako Takeda, Kunie Kohno, Masayuki Yamasaki, Minoru Isomura, Toru Nabika, Atsushi Nagai

**Affiliations:** 1Department of Neurology, Shimane University Faculty of Medicine, Izumo 693-8501, Japan; agaru429@med.shimane-u.ac.jp (G.A.); anagai@med.shimane-u.ac.jp (A.N.); 2Center for Community-Based Healthcare Research and Education (CoHRE), Organization for Research and Academic Information, Shimane University, Izumo 693-8501, Japan; kenta.okuyama@med.lu.se (K.O.); cohre1@med.shimane-u.ac.jp (M.T.); nine_kk@med.shimane-u.ac.jp (K.K.); myamasak@hmn.shimane-u.ac.jp (M.Y.); isomura@hmn.shimane-u.ac.jp (M.I.); nabika@med.shimne-u.ac.jp (T.N.); 3Department of Laboratory Medicine, Shimane University Faculty of Medicine, Izumo 693-8501, Japan; abdullahskm@gmail.com; 4Department of Orthopaedic Surgery, Shimane University Faculty of Medicine, Izumo 693-8501, Japan; yuaorigele@med.shimane-u.ac.jp

**Keywords:** fracture risk, fall, elderly, multimorbidity, logistic regression analysis

## Abstract

Fragility fractures (FFxs), which are a common musculoskeletal injury in older adults, is associated with an increased frequency of falls. Both FFxs and falls may result from drugs, habits, and co-occurring diseases. We aimed to evaluate the effects of various diseases on the risk of FFx. This retrospective study included 1420 individuals aged ≥60 years. We evaluated the history of clinical FFx and diseases using a detailed questionnaire and a health examination. The risk of comorbidities was assessed using the Age-Adjusted Charlson Comorbidity (AAC) Index. We performed binary logistic regression analysis to determine the risk of FFx and falls after adjusting for covariates. In elderly men, the incidence of FFx positively correlated with rheumatoid arthritis and parent’s hip fracture. For elderly women, the incidence of FFx positively correlated with rheumatoid arthritis and antihypertensive drugs but was inversely associated with dyslipidemia and antilipidemic drugs. The FFX risk of older adults with an AAC Index ≥6 was higher than those with an AAC Index of 1–3. In addition, the AAC Index and falls were independently and strongly associated with a higher risk of FFx. Taken together, multimorbidity increases the risk of clinical FFx independent of falls in the community-dwelling elderly population.

## 1. Introduction

Bone fractures are one of the most common musculoskeletal injuries. Evidence of the relationship between lifestyle diseases and fractures has been recently published, and it was reported that the elderly with certain diseases, such as diabetes [1], hypertension [2], chronic obstructive pulmonary disease (COPD) [3], and chronic kidney disease (CKD) [4], are at a higher risk of fracture than the general population. In the elderly, fragility fractures (FFx) affect the quality of life and are associated with increased morbidity and mortality. Various diseases and discomfort have been recognized among the elderly, and, consequently, pharmacotherapy for these diseases can cause public health concerns. Besides steroids [5], the widespread use of drugs, such as sedatives [6], psychotropics [7] and antihypertensives, may increase the risk of falls and fracture. Several other factors can influence bone health, such as lifestyle habits, including physical activity, diet, smoking, and alcohol consumption [8]. In addition, a history of hip fracture and low adherence to antiosteoporosis treatment are associated with a risk of vertebral fracture [9,10]. A family history of hip fractures also appears to be a risk factor for fractures [11,12].

Japan is experiencing the transition to a “super-aging” society in both rural and urban areas. People aged 65 years and older make up a quarter of the total population, and this proportion is estimated to increase to one-third by 2050 [13]. Incapacitating fractures in older individuals decrease their quality of life and impose a considerable financial burden on healthcare services [14] due to impaired mobility, hospitalizations, and the requirement for nursing assistance. FFx results from mechanical forces that normally do not cause fractures are known as low-level or low-energy trauma.

The World Health Organization (WHO) defined low-level trauma as a force that was equivalent to a fall from a standing height or less [15]. FFx occurs more frequently in the elderly. The common sites of FFx include the hip, spine, and wrist. In a recent study using the Japan National Health Insurance Database, FFx was identified in 490,138 of 1,188,754 subjects with bone fractures (41.2%, 345,980 patients/year; 1:4 male-to-female ratio) [16]. The incidence rates of hip fracture did not change among women and significantly increased among men from 2012 to 2015 [17].

It is generally believed that the co-occurrence of multiple diseases, or multimorbidity, and the use of multiple drugs, or polypharmacy, will lead to an increased frequency of falls and the risk of fractures. The Charlson Comorbidity Index (CCI), an effective scale for estimating the mortality of patients with multiple comorbidities, has been used to analyze multimorbidity [18]. Both comorbidities and the FFx risk increase with age and, therefore, we used the age-adjusted Charlson Comorbidity (AAC) index, which has better utility than CCI [19]. Therefore, we conducted this study to understand the current situation of FFx and multimorbidity in a rural area in Japan and to evaluate the effects of multimorbidity on falls and FFx in the general elderly population using the AAC index.

## 2. Materials and Methods

### 2.1. Subjects

This study was undertaken as a part of a cohort study (Shimane CoHRE Study) conducted by the Center for Community-based Health Research and Education, Shimane University. The data were collected from the health examinations of 1720 community-dwelling individuals in Unnan City, Okinoshima-cho, and Ohnan-cho, Shimane Prefecture, Japan, between 2015 and 2016. The inclusion criteria for the study were as follows: (1) individuals who were over 60 years old; (2) individuals who answered the questionnaire regarding the history of bone fracture and history of falls; (3) individuals whose demographic data including age, sex, height, and weight were collected; and (4) individuals who were informed of the protocol and purpose of the study and consented to participate. According to the criteria, a total of 1420 older people (556 men and 864 women) were included in the current study (Figure 1).

### 2.2. Ethics Statement

The study protocol was approved by the institutional ethics committee of Shimane University [#2888], and written informed consent was obtained from all study participants.

### 2.3. History of FFx and Falls

The history of clinical FFx and falls was assessed using a self-reported questionnaire. Participants were asked the following question: “(1) Have you fallen during the last year: yes or no?” and “(2) Have you experienced bone fractures during the last five years: yes or no?” In addition, the participants were asked about the location, cause, and their situation of the bone fracture. To classify FFx and non-fragility fractures, FFx was defined as a pathological fracture that results from minimal trauma (e.g., a fall from a standing height) or without any identifiable trauma [20]. We have excluded finger fractures from the data collected (Appendix A).

### 2.4. Covariates

Other variables were obtained from the responses in the self-evaluation questionnaires. We inquired about age, sex, height, weight, parents’ hip fracture history (yes/no), alcohol consumption (yes/no); current or former smoker (yes/no); chronic diseases (hypertension, dyslipidemia, diabetes, cerebrovascular disease, chronic heart disease, chronic kidney disease, anemia, insomnia, hyperuricemia, peripheral vascular disease, liver disease, gastrointestinal disorders, endocrine disease, rheumatoid arthritis (RA), cancer, allergic disease, and lumbago: yes, no); taking medication (hypertension, diabetes, and dyslipidemia: yes, no); and age at menopause (normal >45 years; early menopause 40–44 years; premature menopause <40 years) [19]. Other physical conditions that were assessed during the health check were: usual walking speed by 10-m walk (<4.5 s, normal; ≥4.5 s, abnormal) [21]; maximum grip strength (in men: ≥28 kg, normal; <28 kg, abnormal; in women: ≥18 kg, normal; <18 kg, abnormal) [22]; the number of teeth (functional teeth: ≥20, normal; <20, abnormal) [23]; denture use (yes, no); and cognitive impairment assessed by the Cognitive Assessment for Dementia, iPad version (CADi), which consisted of 10 simple questions and is self-administered (CADi score ≥7, normal; <7, abnormal) [24]. These potential confounders are reported to be associated with bone fractures and falls [25].

### 2.5. Assessment of Comorbidities

The critical independent variable of this study was the AAC Index. The original Charlson Comorbidity Index (CCI) incorporates the number and severity of preexisting comorbidities [18]. The AAC score is a weighted measure that incorporates age and different subsets of conditions, including hypertension, dyslipidemia, diabetes, cerebrovascular disease, chronic heart disease, chronic kidney disease, anemia, insomnia, hyperuricemia, peripheral vascular disease, liver disease, gastrointestinal disorders, endocrine disease, RA, and allergic disease. Two points were awarded for cancer treatment. The age at diagnosis was adjusted by calculating each decade after 60 years as one point in the AAC Index. For each decade after 60 years, one point was added until 4 points (the age groups 60–70, 71–80, 81–90, and ≥91 years received 1, 2, 3, and 4 points, respectively). Higher scores correspond to an increased mortality risk. The total score was calculated for each patient; based on the total score, and the study population was classified into four ordinal categories based on the AAC Index: 0, 1–3, 4–5, and ≥ 6 points [18,19]. We classified the participants into three groups because none of the participants had an AAC Index of 0.

### 2.6. Statistical Analysis

All data are presented as the mean  ±  standard deviation (SD). The level of significance was set at *p*  <  0.05. The analysis was conducted using SPSS (version 25.0; IBM SPSS, New York, NY, USA). The characteristics of study participants were compared between participants with/without FFx and falls (chi-square test and Fisher’s exact test for categorical data, or Student’s *t*-test for continuous data).

Sex-stratified binary logistic regression models were used to explore the association between the history of FFx and falls (dependent variable), physical characteristics, and covariates (independent variables), and was expressed as odds ratios (OR) and 95% confidence intervals (CI).

The risk rates of FFx and falls among the different AAC groups were compared using the Pearson chi-square test. Binary logistic regression models were used to explore the association between the history of FFx, falls (dependent variable), and the AAC Index, sex, body mass index (BMI), smoking, alcohol consumption, cognitive impairment, number of teeth, denture use, reduced maximum grip strength, reduced usual walking speed, lumbago, parents’ hip fracture history, medications for hypertension, medications for diabetes, and medications for dyslipidemia (independent variables). According to these models, the adjusted odds ratios (ORs) and 95% confidence intervals of FFx and falls were calculated.

## 3. Results

### 3.1. Clinical Characteristics of the Study Population

The characteristics of 1420 participants aged 60 years and older (556 men and 864 women) are shown in Table 1 and Appendix A. Of the study participants, 27 (5%) men and 132 (15%) women experienced clinical FFx during the last five years, and 78 (14%) men and 130 (15%) women experienced falls during the last year. The chi-square test showed a significant difference in the incidence of FFx in both men and women with RA, suggesting that elderly men and women with RA have a higher risk of FFx (Table 1). In addition, a significant difference in the incidence of FFx among three groups of AAC index was found only in women (Table 1).

Compared to women without FFx, women with FFx had a higher ratio of insomnia. However, the ratio of elderly women with dyslipidemia or taking antilipidemic drugs was notably lower in those with FFx than in those without FFx. The ratio of women with fewer functional teeth and denture use was significantly higher in the group with FFx than in the group without FFx. Furthermore, the results suggested that reduced usual walking speed and increased frequency of falls conferred a higher risk of FFx in elderly women. Subjects with FFx had a higher ratio of hip fracture history in their parents.

### 3.2. Association between FFx or Falls and Covariates

A binary logistic regression analysis indicated that RA was associated with a higher OR for FFx in elderly men (Table 2). In elderly women, RA and taking medication for hypertension showed a higher OR for FFx, whereas dyslipidemia or pharmacotherapy for dyslipidemia showed a lower OR for FFx, suggesting that dyslipidemia and antilipidemic treatment might be protective against FFx, whereas pharmacotherapy for hypertension could be an independent risk factor for FFx in women (Table 2). Older women with high AAC index (≥6) had an increased risk of FFx, compared to those with AAC index (1–3). Parents’ hip fracture history showed a higher OR for FFx, especially in men. In addition, falls were strongly associated with FFx in elderly women (Table 2). On the other hand, elderly men with cognitive impairment showed a significantly higher OR for falls (Appendix A). In women, fewer functional teeth were related to a risk for falls.

### 3.3. AAC and Potential Risk Factors for FFx

Next, we examined the cumulative effects of various diseases on clinical FFx and falls using the AAC Index. Compared to those with an AAC Index of 1–3, participants with a higher score (AAC Index ≥6) experienced significantly higher frequencies of FFx in the last five years (Table 3). However, there was no significant difference in falls among the three groups (Table 3).

In the binary logistic regression analysis adjusted for covariates, participants with an AAC Index ≥6 had a significantly higher risk of FFx than those with an AAC Index of 1–3 (adjusted OR: 1.77; 95% CI: 1.14–2.73) independent of falls (Table 4). In addition, participants with an AAC Index of 4–5 had a tendency toward an increased risk of FFx. Furthermore, we found that elderly women had a significantly higher risk of FFx than elderly men (adjusted HR: 3.49; 95% CI: 2.27–5.37) and that the experience of falls during the previous year conferred a high risk of FFx (adjusted OR: 2.16; 95% CI: 1.42–3.28). Also, parents’ hip fracture history was a risk of FFx (adjusted OR: 2.10; 95% CI: 1.32–3.34).

As shown in the additional analysis, it is more likely to find falls in habitual drinkers (adjusted OR: 1.61; 95% CI: 1.13–2.30; Appendix A). Older people with fewer teeth had a higher risk of falls (adjusted OR: 1.65; 95% CI: 1.08–2.52).

## 4. Discussion

FFx and falls are frequent events in the elderly population. However, because of its multifactorial causation, it is difficult to establish a single risk factor for the occurrence of FFx and falls. Thus, the risk of FFx and falls are associated with many factors, including demographics, physical function, diseases, drugs, habits, and the occurrence of self-reported comorbidities. In this cross-sectional study, using binary logistic regression analysis, we found that a higher AAC score leads to a higher risk of clinical FFx in the community-dwelling elderly population. Age, AAC index, falls, parent’s hip fracture history, RA, and antihypertensive drugs were independent risk factors for FFx, although dyslipidemia and antilipidemic drugs were protective factors for FFx in women. In men, we found RA and parents’ hip fracture history as an independent risk factor for FFx. In addition, the risk of FFx in elderly women is likely to be much higher than the FFx risk in elderly men. Furthermore, through binary logistic regression analysis, we found that habitual drinking is an independent risk factor for falls in the elderly. Meanwhile, falls in men are associated with cognitive impairment.

In the present study, we evaluated the risk of clinical FFx and falls in different AAC groups. In the elderly population, a higher AAC Index and history of a parent’s hip fracture suggested a higher risk of FFx. However, the impact of the AAC Index on falls was not significant. In general, the loss of cancellous bone in both men and women begins in the third decade of life and further accelerates with menopause in women [26]. Later, because of menopause or deficiency of sex steroids, cortical bone loss is accompanied by increased porosity [27], thereby increasing the risk of FFx. Diabetes and cardiovascular disease, peripheral arterial disease, and abdominal aortic calcification are associated with a two- to fivefold increased risk of hip fractures [28,29,30]. Decreased kidney function is related to abnormal bone and mineral metabolism that predisposes patients to fractures [4]. The incidence of hip fractures among dialysis patients is 17.4-fold higher than that in the general population [31]. A family history of hip fractures is well known as a risk of hip fractures [32]. In this regard, our results were consistent with the previous findings. Taken together, elderly people with multimorbidity are at risk of fractures, indicating the importance of multimorbidity as a contributory factor in the increased risk of FFx.

Another result was that the immobility resulting from RA-induced muscle pain, weakness, and swelling may increase the risk of falls to some extent [33,34] and increase the incidence of bone fracture. Furthermore, chronic inflammation and glucocorticoid use may be involved in bone fragility among patients with RA. Therefore, together with other studies [35,36], the results of our study demonstrated that patients with RA are at a higher risk of osteoporotic fractures than individuals with non-RA morbidity.

Next, our results showed that antilipidemic drugs can reduce the risk of FFx. Mundy et al. [37] reported that statins might be beneficial for increased bone formation in rodent models, along with inducing some inhibition of osteoclastic activity. Indeed, some reports have shown that antilipidemic medication is negatively associated with fractures in women, which is consistent with the results of our study [37,38]. In contrast, some studies reported that high serum cholesterol levels are associated with fractures [39].

Hypertension is associated with reduced bone mineral density and increased fracture risk [2]. As hypertension is a common and chronic disease, antihypertensive drugs are one of the most widely prescribed drugs. A previous report showed that the blood pressure-lowering effect of all types of antihypertensive drugs per se most probably leads to a decreased calcium loss in the urine, and thus reduce the risk of fractures through decreased loss of bone mineral density [40]. In a pooled analysis of observational studies, treatment with beta-blockers reduced the risk of fractures [41]. In contrast, diuretics are associated with an increased risk of falls [42] and fractures [43]. Our findings showed that taking antihypertensive drugs, but not the presence of hypertension, increased the risk of FFx, suggesting that severe hypertension necessitating medical treatment may increase the risk for FFx in the general population and that the impact of severe hypertension on the risk of FFx is most probably much stronger than the beneficial effects of antihypertensive drugs on the bone.

Falls are the main cause of morbidity and mortality in the elderly [44]. As women become older, their FFx risk increases, which is in agreement with the results of previous studies [45]. The lifetime risk of osteoporotic fractures in women is between 40% and 50%, whereas, in men, it is between 13% and 22% [46]. Falls account for 87% of all fractures in the elderly [25]. The literature shows that the risk of falls increases in elderly people with cognitive impairment, particularly confusion, impaired orientation, and misperception of functional ability [47]. This is consistent with our findings that elderly men with cognitive impairment are more likely to fall. However, a successful strategy for preventing falls in frail elderly individuals with cognitive impairment has not yet been established [48]. Besides these risk factors, alcohol use could be a predictive factor for falls. In a longitudinal study conducted in four communities in the United States, Mukamal et al. analyzed 5841 elderly people and found that drinking more than 14 drinks per week increased the risk of falls by 25% [49]. Therefore, the drinking habits of the elderly cannot be ignored.

In the present study, loss of teeth was associated with falls, especially in women but not with a bone fracture. The most common reason for tooth extraction in the elderly is periodontitis [50]. The postmenopausal state increases the severity of periodontitis by 30% [51]; in addition, the prevalence of osteoporosis in postmenopausal women is as high as 50% [52]. Thus, osteoporosis is expected to accelerate the process of bone loss in chronic periodontitis. Moreover, we found significant association between loss of teeth and sarcopenia in our very recent study [53]. Taken together, further study is necessary to elucidate whether periodontitis or loss of teeth may increase the risk of falls and fractures.

This study has several limitations. First, our research data are relatively small, and the region was quite limited, based on the self-reported data of participants. Second, we collected limited information about the medication use of the study participants. Third, we only analyzed the intersection of FFx and diseases over the past five years and collected clinical fractures with symptoms. Fourth, since participants could not distinguish between osteoarthritis and RA, it is possible that RA in this study contains osteoarthritis and RA-related diseases. Despite these limitations, our study sheds light on the impact of multimorbidity, disease accumulation on the risk of clinical FFx in community-dwelling elderly individuals. In the future, more prospective studies are necessary to confirm our findings from this cross-sectional study.

## 5. Conclusions

We found that older adults with a high AAC Index independently had an elevated risk of FFx, which suggests that multimorbidity increases the risk of fracture.

## Figures and Tables

**Figure 1 jcm-10-03225-f001:**
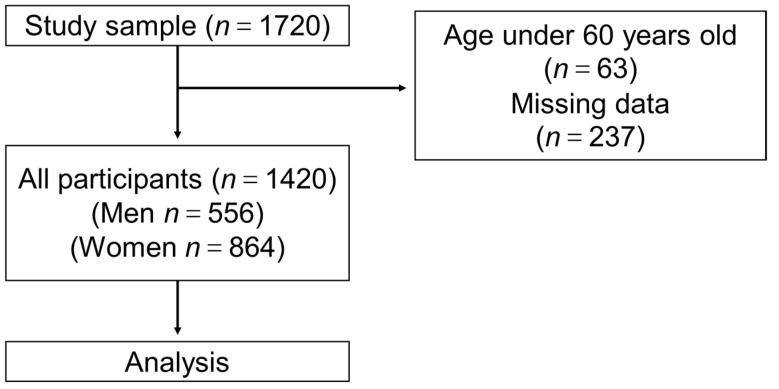
Study flow. Flow diagram of 1420 elderly people who were included in the current study from the Shimane CoHRE study. A detailed explanation is described in Section 2.

**Table 1 jcm-10-03225-t001:** Characteristics of study participants according to fragility fracture during the last five years in men and women.

Variables	Men *n* = 556			Women *n* = 864		
	No FFx	FFx	*p*	No FFx	FFx	*p*
	*n* = 529	*n* = 27		*n* = 732	*n* = 132	
Age, years	71.7 ± 6.5	71.4 ± 6.6	0.90	71.9 ± 6.7	75.3 ± 7.5	<0.01
BMI, kg/m^2^	23.0 ± 2.8	22.0 ± 2.3	0.65	22.7 ± 3.3	22.8 ± 3.3	0.72
AAC Index (1–3)	180 (34.0)	10 (37.0)		245 (33.5)	29 (22.0)	
AAC Index (4–5)	213 (40.3)	9 (33.3)	0.77	283 (38.7)	54 (40.9)	0.02
AAC Index (≥6)	136 (25.7)	8 (29.6)		204 (27.9)	49 (37.1)	
HT	300 (56.7)	13 (48.1)	0.38	409 (55.9)	80 (60.6)	0.31
DM	116 (21.9)	4 (14.8)	0.38	78 (10.7)	17 (12.9)	0.45
DL	187 (35.3)	8 (29.6)	0.54	327 (44.7)	42 (31.8)	<0.01
Stroke	41 (7.8)	1 (3.7)	0.44	23 (3.1)	8 (6.1)	0.10
Heart disease	81 (15.3)	6 (22.2)	0.34	72 (9.8)	19 (14.4)	0.12
Kidney disease	68 (12.9)	4 (14.8)	0.77	34 (4.6)	10 (7.6)	0.16
Anemia	45 (8.5)	3 (11.1)	0.64	158 (21.6)	35 (26.5)	0.21
Insomnia	289 (54.6)	17 (63.0)	0.40	448 (61.2)	98 (74.2)	<0.01
Hyperuricemia gout	50 (9.5)	1 (3.7)	0.31	9 (1.2)	0 (0)	0.20
PAD	10 (1.9)	1 (3.7)	0.51	2 (0.3)	1 (0.8)	0.38
Liver disease	14 (2.6)	0 (0)	0.39	20 (2.7)	4 (3.0)	0.85
GD	42 (7.9)	3 (11.1)	0.56	54 (7.4)	12 (9.1)	0.50
Endocrine disease	7 (1.3)	0 (0)	0.55	47 (6.4)	5 (3.8)	0.24
RA	75 (14.2)	8 (29.6)	0.03	234 (32.0)	67 (50.8)	<0.01
Cancer	49 (9.3)	4 (14.8)	0.34	41 (5.6)	4 (3.0)	0.22
Allergic disease	20 (3.8)	2 (7.4)	0.35	40 (5.5)	7 (5.3)	0.94
smoker	390 (73.7)	22 (81.5)	0.37	35 (4.8)	3 (2.3)	0.20
Alcohol consumption	357 (67.5)	20 (74.1)	0.48	188 (25.7)	29 (22)	0.37
Normal menopause	-	-	-	640 (87.4)	116 (87.9)	0.99
Early menopause	-	-	-	63 (8.6)	11 (8.3)	
Premature menopause	-	-	-	29 (4.0)	5 (3.8)	
Cognitive impairment	84 (15.9)	5 (18.5)	0.72	96 (13.1)	15 (11.4)	0.58
Number of teeth (<20)	212 (42.1)	14 (53.8)	0.24	321 (46.7)	79 (61.7)	<0.01
Denture use (yes)	275 (54.8)	15 (57.7)	0.77	395 (57.6)	87 (68.0)	0.03
Reduced maximum grip strength *	18 (3.4)	1 (3.7)	0.94	51 (7.0)	14 (10.6)	0.15
Reduced usual walking speed **	104 (19.7)	3 (11.1)	0.33	151 (20.6)	47 (35.6)	<0.01
Lumbago	217 (41.0)	8 (29.6)	0.24	320 (43.7)	68 (51.5)	0.10
Parent’s hip fracture history	68 (12.9)	8 (29.6)	0.04	80 (10.9)	22 (16.7)	0.06
Falls	72 (13.6)	6 (22.2)	0.21	94 (12.8)	36 (27.3)	<0.01
Medication for HT	228 (43.1)	11 (40.7)	0.81	305 (41.7)	65 (49.2)	0.10
Medication for DM	98 (18.5)	3 (11.1)	0.33	66 (9.0)	14 (10.6)	0.56
Medication for DL	121 (22.9)	5 (18.5)	0.60	271 (37.0)	37 (28.0)	0.047

**Note:** Statistical analysis was conducted with the paired *t-*test and chi-square test, where *p* < 0.05 was considered as significant. * Reduced maximal grip strength: <28 kg in men and <18 kg in women, ** Reduced usual walking speed by 10-m walk: ≥4.5 s. Abbreviations: FFx, fragility fracture; SD, standard deviation; BMI, body mass index; AAC, Age-Adjusted Charlson Comorbidity Score; HT, hypertension; DM, diabetes mellitus; DL, dyslipidemia; PAD, peripheral artery disease; GD, gastrointestinal disorders; RA, rheumatoid arthritis. Data are presented here as mean ± standard deviation. Data are also presented as a number with a percentage.

**Table 2 jcm-10-03225-t002:** Odds ratios for fragility fracture in men and women by binary logistic regression.

	Men *n* = 556		Women *n* = 864	
Variables	OR (95% CI)	*p*	OR (95% CI)	*p*
Age	1.00 (0.93–1.07)	0.94	1.06 (1.03–1.09)	<0.01
BMI	0.37 (0.05–2.47)	0.30	0.84 (0.41–1.74)	0.64
AAC Index (1–3) (reference)	1	-	1	-
AAC Index (4–5)	0.83 (0.33–2.09)	0.69	1.63 (1.00–2.64)	0.05
AAC Index (≥6)	1.10 (0.42–2.89)	0.84	2.02 (1.23–3.33)	<0.01
HT	0.59 (0.24–1.26)	0.26	1.16 (0.76–1.77)	0.51
DM	0.77 (0.24–2.47)	0.66	1.34 (0.70–2.56)	0.37
DL	0.65 (0.25–1.70)	0.38	0.53 (0.34–0.82)	<0.01
Stroke	0.47 (0.05–4.27)	0.50	1.98 (0.79–4.96)	0.15
Heart disease	1.70 (0.60–4.81)	0.32	1.43 (0.78–2.64)	0.25
Kidney failure	1.11 (0.34–3.67)	0.86	1.13 (0.48–2.66)	0.78
Anemia	1.12 (0.29–4.30)	0.87	1.18 (0.74–1.87)	0.49
Insomnia	1.41 (0.59–3.37)	0.44	1.53 (0.94–2.49)	0.09
Hyperuricemia gout	0.47 (0.06–3.83)	0.48	0.00 (0.00–)	1.00
PAD	2.05 (0.22–19.32)	0.53	7.44 (0.40–137)	0.18
Liver disease	0.00 (0.00–)	1.00	1.02 (0.30–3.47)	0.98
GD	1.18 (0.24–5.84)	0.84	1.22 (0.59–2.53)	0.60
Endocrine disease	0.00 (0.00–)	1.00	0.47 (0.15–1.44)	0.19
RA	3.18 (1.19–8.52)	0.02	1.87 (1.23–2.87)	<0.01
Cancer	1.25 (0.38–4.15)	0.72	0.43 (0.14–1.33)	0.14
Allergic disease	1.88 (0.38–9.36)	0.44	1.27 (0.52–3.13)	0.60
Smoke	1.61 (0.55–4.69)	0.38	0.84 (0.24–2.93)	0.78
Alcohol consumption	1.85 (0.66–5.20)	0.24	1.06 (0.65–1.72)	0.83
Early menopause	-	-	0.82 (0.37–1.82)	0.63
Premature menopause	-	-	0.93 (0.33–2.64)	0.89
Cognitive impairment	1.16 (0.35–3.88)	0.81	0.60 (0.31–1.15)	0.12
Number of teeth (<20)	1.82 (0.57–5.85)	0.32	1.21 (0.68–2.14)	0.52
Denture use (yes)	0.73 (0.23–7.34)	0.59	1.11 (0.63–1.95)	0.72
Reduced maximum grip strength	2.22 (0.23–21.42)	0.49	1.09 (0.54–2.21)	0.82
Reduced usual walking speed	2.13 (0.58–7.79)	0.25	0.64 (0.39–1.05)	0.79
Lumbago	0.60 (0.24–1.52)	0.29	1.10 (0.73–1.67)	0.64
Parent’s hip fracture history	2.81 (1.18–6.69)	0.02	1.63 (0.97–2.73)	0.06
Falls	1.72 (0.63–4.67)	0.29	2.13 (1.31–3.48)	<0.01
Medication for HT	1.00 (0.45–2.22)	0.99	1.50 (1.02–2.21)	0.04
Medication for DM	0.57 (0.16–1.98)	0.38	1.25 (0.67–2.35)	0.49
Medication for DL	0.83 (0.30–2.29)	0.72	0.58 (0.38–0.89)	0.01

**Note:** Statistical analysis was done with binary logistic regression, where *p* < 0.05 was considered to be significant. Abbreviations: OR, odds ratios; CI, confidence interval BMI, body mass index; HT, hypertension; DM, diabetes mellitus; DL, PAD, peripheral artery disease; GD, gastrointestinal disorders; RA, rheumatoid arthritis; DL, dyslipidemia.

**Table 3 jcm-10-03225-t003:** Frequencies of participants with or without FFx and falls in different AAC groups.

AAC Index	No FFx (*n*)	FFx (*n*)	*p-*Value	No Falls (*n*)	Falls (*n*)	*p-*Value
1–3	425	39	0.02	402	62	0.25
4–5	496	63		481	78	
≥6	340	57		329	68	
Total	1261	159		1212	208	

**Note:** Statistical analysis was undertaken with the chi-square test, where *p* < 0.05 was considered to be significant. Abbreviations: AAC, Age-Adjusted Charlson Comorbidity; FFx, fragility fracture.

**Table 4 jcm-10-03225-t004:** Adjusted odds ratio for FFx.

Variables	Adjusted OR (95% CI)	*p-*Value
AAC Index (1–3) (Reference)	1	-
AAC Index (4–5)	1.38 (0.90–2.11)	0.14
AAC Index (≥6)	1.77 (1.14–2.73)	0.01
Men (reference)	1	-
Women	3.49 (2.27–5.37)	<0.01
BMI	0.97 (0.91–1.02)	0.26
Smoke	0.97 (0.49–1.93)	0.93
Alcohol consumption	1.05 (0.70–1.58)	0.81
Cognitive impairment	0.70 (0.41–1.20)	0.20
Number of teeth (<20)	1.54 (0.94–2.50)	0.09
Denture use (yes)	0.99 (0.61–1.61)	0.97
Reduced maximum grip strength	1.05 (0.55–2.00)	0.88
Reduced usual walking speed	0.66 (0.43–1.01)	0.06
Lumbago	1.10 (0.77–1.57)	0.59
Parent’s hip fracture history	2.10 (1.32–3.34)	<0.01
Falls	2.16 (1.42–3.28)	<0.01
Medication for HT	1.35 (0.96–1.90)	0.09
Medication for DM	0.78 (0.46–1.34)	0.38
Medication for DL	0.76 (0.52–1.12)	0.17

**Note:** FFx risk was adjusted with sex, BMI, smoking, alcohol, cognitive impairment, number of teeth, denture use, reduced maximum grip strength, reduces usual walk speed, lumbago, parent’s hip fracture history, falls, medication for HT, medication for DM, and medication for DL. Statistical analysis was done with binary logistic regression, where *p* < 0.05 was considered as significant. Abbreviation: AAC, Age-Adjusted Charlson Comorbidity Score; FFx, fragility fracture; Adjusted OR, adjusted odds ratio; CI, Confidence interval; HT, hypertension; DM, diabetes mellitus; DL, Dyslipidemia.

## Data Availability

The datasets used and/or analyzed during the current study are available from the corresponding author upon reasonable request.

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
