# Peer review of "Effect of Multimorbidity on Fragility Fractures in Community-Dwelling Older Adults: Shimane CoHRE Study"

_jcm, 2021, doi:10.3390/jcm10153225_

Round 1

Reviewer 1 Report

Well-written manuscript

Author Response

Thank you for your careful and constructive review very much.

We editted the revised manuscript including English.

Reviewer 2 Report

This very interesting study is a contribution to great discussion and wide scholarly broadcasting. Aging populations create multiple overlapping medical problems.

Many medical aspects of the health of the elderly population were examined here. These have been evaluated statistically.

I'm not sure the authors are able to assess which of the variables detected are causes and which are effects. I think it is generally very difficult. So I am very supportive of doing such analyses and I think this study should be approved.

Author Response

Thank you for your careful and constructive review very much.

Reviewer 3 Report

  1. Is not clear why authors decided to perform this study. In introduction a potential role of AAC Index in predicting fractures and falls was neither explained nor supported by the literature. In a terse aim of the study provided in introduction (lines 69-71: “we conducted this study to understand the current situation of FFx in Japan and to evaluate the effects of multimorbidity in the general elderly”), the first part of the sequence is unclear, while the second also is unclear and in addition does not match the title of this work. In my opinion the use of AAC Index in prediction of bone fractures and falls is inappropriate. Generally, analyses based solely on binary (yes/no) measures along with overadjustment for confounders (authors used 33 confounders) may either increase net bias or reduce precision without affecting bias. In addition, the AAC Index does not necessarily account for the effects of fine gradations of comorbidity severity that might be reflected in continuous variables and hence, it is possible that continuous measures of diseases and treatments might outperform the Charlson score. Furthermore, the AAC analysis does not capture a wide range of more relevant conditions known to affect the study outcomes but which were not studied in this work, such as low bone mineral density, low bone strength, neurological disorders, muscle diseases including sarcopenia, blurred vision, low vitamin D level, etc.
  2. The evaluation of the impact of antihypertensive treatment on the study outcomes using solely a yes/no measure is of limited value. Among the antihypertensive drugs routinely used in clinical practice, some have the potential to affect bone fractures through bone mineral density (e.g. thiazides [Bone 2020;138:115507]), falls through orthostatic hypotension or bradycardia (e.g. beta-blockers, alfa1-receptor blockers or loop diuretics) or both, while most other drug classes seem to have rather a neutral effect. Therefore, researchers should consider antihypertensive treatments individually.
  3. Regarding hypolipemic drugs, statins have been suggested to reduce fracture risk but to increase the risk of falls [QJM 2009;102(9):625-33]. In addition, due to hypertriglyceridemia many cases are treated with fibrates or Omega-3 drugs, which seem to have a neutral effect on study outcomes. Therefore, authors should consider hypolipemic therapy individually.
  4. Regarding antidiabetic therapy, thiazolidinediones are known to increase fracture rate and decrease bone mineral density [Am J Therapeutics 2020;27(6):e701-e704], while insulin may exert positive effects on bone metabolism but may increase the rate of falls due to hypoglycemia and diabetic neuropathy. Therefore, authors should consider antidiabetic therapy individually.
  5. History of fractures and falls (lines 94-100): why authors asked the participants about prior falls only within the last one year and fractures within five years? Based on these data, how authors diagnosed a fragility fracture defined as a fall from a standing height for example 3 years before evaluation, while the fall questionnaire captured only a one year interval?
  6. Covariates (lines 101-7): there is no information about smoking but in statistical analyses authors calculated ORs adjusted for smoking. Please explain.
  7. Table 1: it is rather unlikely that the percentage of RA patients was so high (e.g. >50% of women with fractures) suggesting that less common RA and common osteoarthritis were not differentiated. This is important issue as both conditions exert different effects on fracture rate.
  8. Table 1: it is rather unlikely that the difference in rates of smokers between genders was so high (73.7-81.5% in males vs. 2.3-4.8% in females). How authors interpret these findings?
  9. Table 1 and Table 3S: what was the general rationale for counting the number of teeth in the study participants? It looks that having less than 20 teeth predispose to fractures and falls (!), and in addition, this is exacerbated by the use of dentures (!!). Humorously, there is possible one explanation: a lower number of teeth in females (but not males) could be a consequence of falls but even so, a negative role of denture use still remains unclear… These findings are not discussed in the manuscript.
  10. Methods (lines 97-98): “The participants were asked about the location, cause, and their situation of the bone fracture”. Regarding hand fracture: how did the respondents differentiate in the questionnaire whether the fracture involved the fingers, metacarpals or wrists? All these fractures can be colloquially called a hand fracture; however, a fragility fracture refers rather to the wrist fracture (Colles fracture) than fractures in other hand localizations.
  11. It is unknown, how authors defined a fragility spine fracture. Did they evaluated lateral spine X-rays?
  12. This study does not provide new data either on fragility fractures or falls.

Author Response

Thank you for your careful and constructive review very much.

We editted the revised manuscript including English.

1), and 12) There are many interactions among organs, and thus lots of disease-disease associations have been reported. Although multimorbidity has been recognized especially in older population, the association of bone fracture with multimorbidity has not been clarified. Therefore, we conducted this study to elucidate the association. This is the first study to evaluate the association between multimorbidity and fragility fractures in older people, even if the AAC index has some shortcomings or limitations as the reviewer raised. According to the reviewer suggestions, the sentence in the Introduction was modified in the revised manuscript.

Therefore, we conducted this study to understand the current situation of FFx and multimorbidity in rural area in Japan and to evaluate the effects of multimorbidity on falls and FFx in the general elderly population using the AAC index.

2), 3), and 4) Thank you for your critical comments. Because we did not collect information about specific medication, we are not able to individually explain, which is our limitation. However, at least we can say that no participant used insulin. In Japan, the prescription rate of thiazolidinediones is very small for diabetic patients. For hypertension, Ca-channel blocker, ACI, and ARB are major medications whereas beta-blocker, alpha-blocker, diuretics are minor.

5) We diagnosed FFx based on the information obtained from the subjects’ questions. It is difficult for us to remember the history of falls for more than 1 year, but the history of fractures within 5 years will be remembered more clearly. For more accurate information, we designed to collect the history of falls within 1 year and the history of fractures within 5 years.

6) Thank you for pointing it out. We added information about smoking.

We inquired about age, sex, height, weight, parent’s hip fracture history (yes/no), alcohol consumption (yes/no), current or former smoker (yes/no); chronic diseases (hypertension, dyslipidemia, diabetes, cerebrovascular disease, chronic heart disease, chronic kidney disease, anemia, insomnia, hyperuricemia, peripheral vascular disease, liver disease, gastrointestinal disorders, endocrine disease, rheumatoid arthritis (RA), cancer, allergic disease, and lumbago: yes, no).

7) Thank you for pointing the critical issue. Since subjects could not accurately distinguish between osteoarthritis and rheumatoid arthritis, it is possible that RA contains osteoarthritis and RA-related disorders. This is the limitation of our research. We added some explanation on this issue in the limitation of the revised manuscript.

Fourth, since participants could not distinguish between osteoarthritis and RA, it is possible that RA in this study contains osteoarthritis and RA-related diseases.

8) We collected information about smokers including current smokers and former smokers. In our study, many people stopped smoking halfway through, but we still collected their information and wanted to compare with never smokers. Current smokers are about 20% in men and 1% in women, which remains big gender difference. However, this may be characteristics in our population and is shown in our previous studies (PLoS One 11(2): e0149452, 2016) (J Atheroscler Thromb 25: 42-54, 2018).

9) Thank you for pointing out an important issue. We added content about the number of teeth in the discussion section as well as the references.

In the present study, loss of teeth was associated with falls especially in women but not with bone fracture. The most common reason for tooth extraction in the elderly is periodontitis [Reich E. Int Dent J 2001]. The postmenopausal state increases the severity of periodontitis by 30% [Albandar JM, et al.  Periodontology 2000]; in addition, the prevalence of osteoporosis in postmenopausal women is as high as 50% [Kanis JA, et al. Osteoporosis international 2008], thus osteoporosis is expected to accelerate the process of bone loss in chronic periodontitis. Moreover, we found significant association between loss of teeth and sarcopenia in our very recent study [Abe T, et al. PLoS One 2021]. Taken together, further study is necessary to elucidate whether periodontitis or loss of teeth may increase the risk of falls and fractures.

Reich, E. Trends in caries and periodontal health epidemiology in Europe. Int Dent J. 2001, 51, 6 Suppl 1, 392-8.

Albandar, J.M.; Rams, T.E. Global epidemiology of periodontal diseases: an overview. Periodontol 2000. 2002, 29, 7-10.

Kanis, J.A.; Burlet, N.; Cooper, C.; Delmas, P.D.; Reginster, J.Y.; Borgstrom, F.; Rizzoli, R.; European Society for Clinical and Economic Aspects of Osteoporosis and Osteoarthritis. European guidance for the diagnosis and management of osteoporosis in postmenopausal women. Osteoporos Int. 2008, 19, 4, 399-428.

Abe T, Tominaga K, Ando Y, Toyama Y, Takeda M, Yamasaki M, Okuyama K, Hamano T, Isomura M, Nabika T, Yano S: Number of teeth and masticatory function are associated with sarcopenia and diabetes mellitus status among community-dwelling older adults: A Shimane CoHRE study. PLoS One. 2021, 16(6), e0252625.

10) Because we have excluded finger fractures from the data we collected, we now change the name of hand fracture to wrist fracture in Table S3. We added the sentence in the method as well.

We have excluded finger fractures from the data collected.

11) Our information about fractures is based on the questionnaire. Thus, we only collected clinical fractures of the spine with back pain or lumbago. Although we did not check X-rays of the spine, medical doctors should make diagnosis according to lateral spine X-rays or MRI. Since we missed asymptomatic fragility spine fractures, the prevalence of total fragility fractures is most probably much higher than what we have showed. We added some explanation on this issue in the limitation of the revised manuscript.

Third, we only analyzed the intersection of FFx and diseases over the past 5 years, and collected clinical fractures with symptoms.

Round 2

Reviewer 3 Report

Thank you for responding to the comments. However, the changes made in the revised version are mainly cosmetic and do not address the main methodological concerns. Again, first, the questionnaires were too general which made it impossible to collect more detailed information. If information on treatment has not been collected, how do the authors know, for example, that none of the respondents took insulin? Second, attempts to explain that periodontal disease is more common in osteoporosis do not apply because the authors have not studied osteoporosis at all. Thirdly, the AAC index is a good tool for assessing outcomes such as overall mortality, DALYs or quality of life, but it is not suitable for assessing the frequency of specific symptoms such as falls or fractures, which the authors were not able to adequately define based on the available data.  Since AAC is only a measure of overall health, the conclusion that AAC contributes to falls or fractures is obvious, and the current work adds no new data to it. In this way, many other relationships can be demonstrated, e.g. the worse the AAC result, the worse the ability to solve crossword puzzles or the greater the frequency of depression. But does it make sense?

Author Response

We appreciate your careful and precious review very much. I understand the meaning of your comment and our limitation in this study. Therefore, we modified the manuscript and added some limitation in the revised manuscript, although you might feel cosmetic.

You stated that “Since AAC is only a measure of overall health, the conclusion that AAC contributes to falls or fractures is obvious.” That is probably true. However, we do not know and need to know whether overall health (maybe similar to multimorbidity) is associated with falls and fractures. Thus, we conducted this study and found that the AAC is a good tool for the risk assessment of falls and fractures in the present study.

As for the first question, we know well our participants and their medical problems in the community. That’s why I said that no participant used insulin. However, we do not have detailed information about all medicine prescribed, thus we could not address medication in our analysis of the relationship between bone fracture and multimorbidity.

Thank you again for your review and constructive comment.